# Daily Time Use by Activity of Community-Dwelling Older Koreans: Focus on Health Management

**DOI:** 10.3390/ijerph18041688

**Published:** 2021-02-10

**Authors:** Hana Ko

**Affiliations:** College of Nursing, Gachon University, 191 Hambakmoero, Yeonsu-gu, Incheon 21936, Korea; hanago11@gachon.ac.kr; Tel.: +82-32-820-4210

**Keywords:** health status, senior centers, time management, aged

## Abstract

This study aimed to examine the daily time use by activity and identified factors related to health management time (HMT) use among 195 older adults (mean age = 77.5, SD = 6.28 years; 70.8% women) attending a Korean senior center. Descriptive statistics were analyzed and gamma regression analyses were performed. Participants used the most time on rest, followed by leisure, health management, daily living activities, and work. The mean duration of HMT was 205.38 min/day. The mean score for the subjective evaluation of health management (SEHM) was 13.62 and the importance score for SEHM was 4.72. Factors influencing HMT included exercise, number of chronic conditions, fasting blood sugar level, low density lipoprotein level, and cognitive function. HMT and frailty significantly predicted SEHM. HMT interventions focus on promoting exercise and acquiring health information to improve health outcomes among older adults in senior centers.

## 1. Introduction

Time is a limited and finite resource. It is subjective, closely related to individuals’ daily lives, and is an essential resource for activities. Therefore, how individuals spend their time is an indicator of life satisfaction and quality of life; it influences success in seeking meaningful activities [1]. 

Time use varies according to sex, age, economic capacity, and health. Therefore, understanding how older adults spend their days is important in identifying their problems and desires [2]. Older adults experience sociopsychological events, such as retirement and widowhood and physical changes, including muscle weakness and function loss [3,4,5]. This discourages them from engaging in external activities, and the content and quality of their leisure activities get simplified, leading to unproductive use of time [6]. In the Korean Life Time Use Survey 2019 [7], the use of time for sleep or meals, which accounts for most of the time use of the older adults, increased by 24 min to 730 min from five years ago, but leisure time such as participation in social activities was 441 min, decreased by 25 min from the 2014. World Health Organization [8] defines health aging as the process of developing and maintaining the functional ability that enables well-being in older age. Therefore, it is emphasized to maintain and improve the functional health of the older adults [8], and studies related to frailty, a complex concept of physical, cognitive, and psychosocial, are being conducted [9]. Therefore, in terms of social health, it is necessary to examine the use of time by the older adults, including frailty.

Some studies involving the use of time in older adults [1,2,6] have analyzed their daily routines using occupational questionnaires [10] or the Life Time Use Survey [7]. However, these studies examined only general characteristics and the time spent on work, family, and leisure, and the questionnaires were designed for the overall population without considering the characteristics of older adults. Most studies examining the actual use of time involved mainly groups with specific diseases or disabilities [6,11,12]. The analysis of time use did not include subcategories concerning health, and information regarding health management time (HMT) use among older adults was insufficient. Consequently, there is a lack of knowledge regarding older adults’ health needs and health problems. 

Time-use surveys are now part of the health surveillance programs of several countries, but current population statistics may have misclassified older adults as being physically inactive because of narrow focus and may fail to characterize the populations most appropriate to target [13]. The Korean government has accumulated statistical data regarding public time use via a national daily time management survey since 1995. In the Korean Life Time Use Survey [7], women (105 min) exhibited greater engagement in self-care practices than men (97 min). However, HMT was not measured precisely. As interest in health management continues to increase, it is necessary to examine time use in greater depth to determine how functional older adults spend time on health management. 

The implications of a subjective evaluation of time use are important. This was not considered in the Korean Life Time Use Survey [7], which failed to analyze time use according to health status or include subjective evaluation of time management. However, an individual’s subjective evaluation of his/her activities is an important criterion in determining whether the current situation should be improved or maintained. Therefore, it is necessary to include individuals’ subjective evaluation of their current time use patterns, including how well time is spent, how important the activity is, and how much it is appreciated.

Various studies have measured health management using the objective indicator “health activity” and analyzed it in relation to general characteristics [1,2,6]. Moreover, recent studies have focused on health activity or health management patterns according to health status [14,15], as health management prevents health deterioration in older adults and contributes to early detection and delay. In general, older adults are known to care about health issues [14]; however, few studies have examined the time spent on health management as a fundamental factor in successful aging. It is anticipated that the time required for health management—including doctor’s appointments, health checks, medication, and activities to maintain or improve health (e.g., exercise)—increases with age.

According to Newman [16], the expansion of consciousness is a process involving health management, and this can be measured according to time, space, and movement. He posited that the perception of time is an indicator of health status, and movement through space is essential for the development of the concept of time, which is included in the measurement of time as an indication of health. Therefore, it is necessary to examine the meaning of life of the elderly through not only measuring health care time but also subjective evaluation of time management. The use of time for simplified activities prevents engagement in other activities, which could cause problems with regard to health and quality of life [17]. The use of time for health management influences engagement in meaningful activities. Therefore, it is necessary to analyze daily time usage patterns in older adults, determine the amount of time spent daily on health management, and identify the factors that influence these patterns. Further, it is likely that data collection regarding time use depends on human memory, and errors could occur in the arbitrary assignment of time to two or more activities with the same purpose (which should belong to the same category) as individual activities. Therefore, response bias, whereby measured values are lower relative to actual values, can occur [18]. The generalized gamma model (GGM) is a statistical approach for controlling response bias in subjective evaluation (such as that of HMT) that involves a positively skewed distribution and shows heteroscedasticity [19]. It is necessary to apply GGM to the time analysis of the older adults, as in studies that used GGM to investigate nursing activity time.

In this light, this study examined daily time use, analyzed the specific activities performed by older adults for health management, and identified influential factors. The study was conducted to provide basic data for the development of effective programs and health management activities for older adults attending senior centers. It aimed to (1) understand older adults’ general characteristics and subjective evaluation of time management, (2) explore health management activities, and the differences of HMT in relation to health status, and (3) identify factors influencing HMT and SEHM.

## 2. Materials and Methods

I conducted an analytical study that examined the activity domains in daily time use among older adults attending a senior center and identified HMT characteristics and related factors.

### 2.1. Participants and Data Collection

Adults aged ≥ 65 years were recruited from a senior center in Seoul between March 2015 and May 2016 using convenience sampling. The criteria for inclusion were the ability to communicate verbally, the ability to understand the study objective and questionnaire content, and the provision of consent for study participation. The sample size required for multiple regression analysis was calculated using G*Power 3.1 (Heinrich-Heine-University, Düsseldorf, Germany), with a significance level of 0.05, explanatory power of 0.95, and an intermediate effect size of 0.15. The required sample size for regression analysis with 10 independent variables was 172. Considering possible dropouts, we recruited 200 subjects. Of the 200 questionnaires completed by the individuals who agreed to participate in the study, 195 were analyzed, and the remaining five were excluded from the analysis because of participants’ withdrawal from the study or missing data.

Data were collected from older adults who regularly attended the senior center and were newly registered as members. The questionnaire was distributed to older adults who fulfilled the inclusion criteria. Most subjects completed the questionnaire themselves. However, for those who were unable to do so, a study assistant, who had completed training for study data collection, read the questionnaire aloud and recorded the subject’s responses. The gerontological nurse practitioner (GNP) evaluated frailty, cognitive function, and depression in face-to-face interviews to examine the participants’ overall health status. 

### 2.2. Measures

#### 2.2.1. Time Management

The Occupational Questionnaire developed by Smith et al. [10] was used to collect data regarding time management. The scale measures daily activity in 30-min units of time and involves behavior categorization and subjective evaluation. Following the behavior categorization, subjective evaluation of each behavior (which involved competency, importance, and enjoyment) was measured using a five-point Likert scale, with total scores ranging from 0 to 15 and higher scores indicating higher satisfaction of the activity.

#### 2.2.2. Behavioral Categorization

Based on the results of the Life Time Use Survey [7] and studies using occupational questionnaires for older adults [1,14,17], 20 mid-level categories and 56 low-level categories were classified into four high-level categories, as follows: work (employed or self-employed, unpaid work for family, agricultural and fishery work for self-consumption, and seeking employment), daily living (personal maintenance, housekeeping, cleaning and organization, household chores, and caring for family), leisure (socializing, using media, religious activities, hobbies, participation in activities, and voluntary work), and rest (sleeping). Originally, the Life Time Use Survey [7] included personal health management, self-treatment, medical services, and resting when ill as a single category. However, in the current study, personal health management was classified into four categories based on the participants’ responses to open-ended questions: self-treatment (medication, heat therapy, and home remedies), health services (medical services at the clinic and receipt of health services), physical activity (physical therapy, rehabilitation exercises, and exercises to maintain health), and other health-related activities (obtaining health information, health checks, and resting when ill).

#### 2.2.3. Nutritional Status

The Determine Your Nutritional Health checklist [20] measures nutrition management in older adults. The questionnaire includes 10 questions that measure nutrition management during the preceding month, with responses provided using a binary scale (yes =1, no = 0). Overall scores are classified as good (0–2 points), moderate risk (3–5 points), and high risk (6 points or more). The reliability and validity of this tool are secured in the older adults, so it was used in the National survey of older Koreans [21] and recently study [9].

#### 2.2.4. Depression

The Korean version of the Geriatric Depression Scale [22] was used to measure depression levels. The scale measures emotional discomfort, negative thoughts, unhappiness, physical frailty, reductions in energy, cognitive function problems, social attention, and reductions in activity during the preceding week. It consists of 15 questions, with responses provided using a binary scale (yes = 1, no = 0). Total scores range from 0 to 15, and scores of 0–4, 5–9, and 10–15 indicate normal state, moderate depression, and depression, respectively. The reliability and validity of this tool are secured in the older adults, so it was used in the National survey of older Koreans [21], and its reliability was Cronbach’s alpha was 0.88 at time of development [22], 0.89 in recent study [9], and 0.80 in the present study.

#### 2.2.5. Frailty

Frailty was measured using the Korean Frailty Index [23]. The questionnaire consists of eight items pertaining to general health status, medication, nutritional status, emotional status, urinary incontinence, walking ability, and communication problems, with responses provided using a binary scale (yes = 1, no = 0). Total scores range from 0 to 8, and scores of 0–2, 3–4, and ≥5 indicate robust condition, pre-frailty, and frailty, respectively. The reliability of Cronbach’s alpha in previous studies in the older adults was 0.65 [23], and in this study, it was 0.62.

#### 2.2.6. Cognitive Function

Cognitive function was measured using the Korean version of the mini-mental state examination [24]. The total score is 30 points, with higher scores indicating better cognitive function. This tool is widely used as a screening tool for dementia for the older adults in the community due to its simplicity, high sensitivity, and specificity [25]. The reliability of Cronbach’s alpha in previous studies in the older adults was 0.87 [25], and in this study, it was 0.89.

#### 2.2.7. Physical Examination

Physical examinations were performed by the researchers who were certified GNP. The current health status was examined by measuring blood pressure, glycated hemoglobin (NycoCard Reader II, Axis-Shield PoC, Rodelokka, Oslo, Norway), fasting blood sugar (FBS), and low-density lipoprotein (LDL) levels (Cholestech L.D.X. Anaylzer, Cholestech Corporation, Hayward, CA, USA). 

### 2.3. Ethical Considerations

The study plan was reviewed and approved by the Institutional Review Board of the affiliated Seoul National University (IRB No. 1503/001-005). Prior to data collection, the subjects were informed that their data would be used only for the study’s purposes and that they could withdraw from the study at any time without incurring any disadvantage. A written explanation of the study was provided, and written informed consent was obtained from all subjects. The GNP assessed the subjects’ safety during the health examination and questionnaire completion. 

### 2.4. Data Analysis 

Data were analyzed using IBM SPSS Statistics 23.0 (IBM Corp., Armonk, NY, USA). In particular, the GGM was used to control for potential response bias [26]. This is a useful means of controlling for estimated values that differ from real values when data have a positively skewed distribution, are close to mean values, include missing values, or estimate mean values using log transformation values such as those obtained via time analysis [18]. Means and standard deviations were calculated for general characteristics, health status, and time use. After normality test with the Shapiro–Wilk test, relationships between health status, and HMT were analyzed using Kruskal–Wallis tests. The Mann Whitney U test were performed as post-hoc analyses; and factors influencing HMT and SEHM and the extent of their influence were analyzed using GGM.

## 3. Results

### 3.1. General Characteristics of Participants

Of the 195 enrolled participants, approximately 39.0% were older than 80 years; 70.8% were women; 17.9% were employed; and 45.1% lived alone. The proportion of subjects who finished elementary school was the largest, and 44.1% reported middle socioeconomic status. The cognitive level of most participants was normal (87.2%), but there were also participants with severe cognitive impairment (1.5%; Table 1).

### 3.2. Subjective Evaluation of Time Management

The score for subjective evaluation of work was the highest, followed by health management (Table 2). With respect to SEHM, the competence score was lower than the importance score. The average daily duration of HMT was shorter relative to the mean durations of resting and leisure time and longer relative to the duration of time spent on daily living activities and work.

Table 3 shows the types of health management activities. The duration of time spent on medication administration in the self-treatment category, orthodox medical service use in the health services category, exercise in the physical activity category, and resting when ill in the other health-related activities category were higher relative to those for the other activities in each category.

### 3.3. Health Status and Health Management Time

The findings revealed more HMT specificity in health status (Table 4). In particular, patients with chronic disease, medication, exercise, fasting glucose, HbA1C, lipid, and the nutritional status of older adults showed statistically significant differences in HMT.

### 3.4. Factors Influencing Health Management Time and Subjective Evaluation of Health Management

GGM analyses included the number of chronic conditions, FBS, LDL level, and cognitive function (which were significant factors in the HMT–health status correlation analysis), number of medications, nutrition score, and exercise (which differed significantly according to health characteristics), SEHM score, frailty, health time use, and number of chronic conditions (Table 5). 

The likelihood ratio chi-square result for the HMT model with seven variables was 40.61 in the GGM analysis (*p* < 0.001), and the total deviance was 45.47, which was lower than the degrees of freedom 186; therefore, the model was reliable. HMT was significantly related to no exercise, number of chronic conditions, FBS, LDL, and cognitive function, while SEHM was significantly related to the participants’ HMT and frailty. Meanwhile, the likelihood ratio chi-square result for the SEHM model with three variables was 21.48 (*p* < 0.001), and the total deviance was 3.85, which was lower than the degrees of freedom 189.

## 4. Discussion

This study examined daily time use and health status of older adults attending a senior center, determined the amount of time spent on health management, and identified patterns and related factors. Examination of older adults’ health status and evaluation of their daily time use provided basic data for designing suitable programs and for establishing health activity plans that could contribute to effective time management for older adults attending senior centers. Therefore, the findings could ultimately help older adults use their time in a healthy and effective manner to achieve successful aging.

In this study, the HMT of older adults was shorter than the mean durations of resting and leisure time and longer relative to the duration of time spent on daily living activities and work. However, in the Report on the South Korean National Life Time Use Survey [7], HMT was the lowest compared with time spent on other activities. This result can be attributed to the behavior classification survey being systematically subdivided according to the older adults’ characteristic. The “health management activities” category was a high-level category, whereas the “exercise to maintain health” and “rehabilitation exercise” categories were mid-level categories, similar to “self-treatment,” “medical services,” and “other health related activity,” and the analysis was based on real responses. Moreover, according to the guidelines for senior center projects established by the Ministry of Health and Welfare [27], support for older adults to live healthier lives should be provided; functional recovery support is part of general support, and physical therapy, rehabilitation exercise, and health education projects should be implemented as part of health improvement support. In accordance with the guidelines, the senior center attended by the subjects offered 2 h of regular exercise time for all members every morning. It also has a well-established physical therapy program, which was included as a health management activity; this could also explain the longer duration of HMT in the current study. 

A previous study examining senior center programs [28] found that health management (88.9%) and health exercise (86.7%) were the most frequently implemented programs at senior centers. In addition, older adults identified health education as the most important program (43.2%), and health management and exercise, the most helpful programs. According to the National Assembly Research Service [29], health-related services should involve programs that older adults prefer and consider the most helpful. Health programs at senior centers directly influence members’ health management. The World Health Organization [30] emphasized that it is important for age-friendly cities to increase older adults’ social relationships and life satisfaction. Therefore, in consideration of the increasing number of senior centers and users, the direction of health projects at senior centers requires clarification to ensure that the facilities provide primary health management.

Interestingly, the SEHM results were high relative to those for work and other activities, regardless of objective time use. The importance score was particularly high, and participants considered health management a meaningful activity and made an effort to perform such meaningful activities. However, the evaluation of competency in one’s own health management was lower than that for importance, which indicates the necessity of training and education in health management for older adults. Because the promotion of a healthy lifestyle for older adults with chronic diseases is affected by self-esteem, self-efficacy, and social support [31], an effective time management program that strengthens these competencies is necessary, through which confidence and life satisfaction should be increased. 

The concept of health management is complex and cannot be explained by factors measured using only behavior and knowledge. Therefore, this study examined the concept of health management using objective indicators, such as the subjects’ actual activities, time use, and subjective evaluation. HMT in older adults is very important, and its value lies in enabling older adults to recognize the importance of exercise, chronic conditions, and frailty, which influence health management. The findings of a previous study on health status [32] showed significant relationships between chronic conditions, need for medication, exercise, and poor health status. Regarding HMT significant difference was observed for participants with chronic diseases in terms of number of medications, regular exercise, fasting glucose, lipid level, and nutritional status. These findings indicate that the condition of older adults with chronic diseases deteriorates, their lifestyles are more evidently affected, and their demands increase in various ways. Thus, it is necessary to assess the needs of older adults according to the stage of disease and to develop effective interventions. The transtheoretical model (TTM) is the most comprehensive and integrated model of behavioral changes among older adults’ health behavioral interventions. It considers TTM and its constructs as a useful tool in the process of creating, developing, and evaluating tailored interventions to promote practice of physical activity, including exercise [33]. Based on this model, health behavior interventions for older adults are needed to prevent chronic disease risk, improve mental capacity, and maintain cognitive functions.

The results of previous studies indicate that regular exercise, blood sugar levels, and metabolic syndrome are significant factors affecting mortality [32,34]; health management is related to successful aging [35]; and efficient time use intervention reduces depression and increases self-esteem and satisfaction with life in older adults [1]. Bearing in mind these findings, it is necessary to develop interventions that support efficient time management while considering health-related factors for older adults. This could improve the quality of life of older adults and contribute to successful aging. 

Meanwhile, the proportion of participants diagnosed with diabetes in the current study was 32.9%, and diabetes continued to rank in the top five causes of death for older adults in South Korea [36]. Although senior centers are classified as leisure welfare facilities, older adults in these centers have health problems. Therefore, as the number of senior centers continues to increase [36], the number of community-dwelling patients with chronic diseases, such as diabetes, is also expected to continue to increase. Diabetes requires continuous self-management, which includes symptom management, changes to living habits, management of lipids, blood pressure, cardiovascular risk, and medication, to ensure independent living [37]. Clearly, there is a strong need for health management. As such, it is necessary to develop comprehensive and tailored interventions that include the variables found to be significant in this study, to ensure that older adults with diabetes use their HMT effectively.

Frailty is a multidimensional concept [38] and recently, the concept of “social frailty” has been increasingly emphasized in relation to older adults because social frailty concerns mood, nutrition, muscle weakness, cognitive decrease, and can lead to disability and mortality [38,39]. In this study, frailty was confirmed to have a negative effect on SEHM. Thus, it is also necessary to promote health programs designed specifically for community-dwelling, frail, older adults who do not have disabilities in daily functioning but have difficulty in leading an active life. Evidence is accumulating regarding interventions for frail older adults, including an integrated chronic care intervention with different frailty levels. A systematic protocol study is also being attempted [40]. Furthermore, as in a previous study [12] in which time intervention for stroke patients was found to be effective in improving life satisfaction and self-esteem, further studies of time use intervention that leads to improvement of SEMT, including life satisfaction of older adults, are needed.

## 5. Conclusions

Health management accounts for a large proportion of older adults’ daily time use, and they considered it important. Older adults attending the senior center had little or no functional disorder, but they exhibited risk factors and had been diagnosed with chronic conditions. HMT depends on a person’s health status; therefore, interventions should be tailored as such. This study is noteworthy because its findings indicate that exercise and health education should be included in center programs for older adults to strengthen their competence and prevent frailty by promoting regular exercise. In addition, this study examined the significance of the older adults’ health management by linking the health theory with the use of time as consciousness expansion. However, this study has limitations. Specifically, it investigated the use of time by older adults before the COVID-19 pandemic; thus, it is necessary for longitudinal research to investigate how older adults are managing their health and time use in the current situation.

## Figures and Tables

**Table 1 ijerph-18-01688-t001:** Participants’ general characteristics (*N* = 195).

Variables		*N* (%)
Age, years	Mean (SD)	77.54 (6.28)
Gender	Men	57 (29.2)
	Women	138 (70.8)
Employment status	Employed	35 (17.9)
	Unemployed	160 (82.1)
Living arrangements	Alone	88 (45.1)
	With spouse	66 (33.8)
	Two generations	28 (14.4)
	Three generations	13 (6.7)
Education level	None	40 (20.5)
	Elementary school	57 (29.2)
	Middle school	24 (12.3)
	High school	47 (24.1)
	≥College	27 (13.8)
Socioeconomic status	High	4 (2.1)
	Middle	86 (44.1)
	Low	105 (53.8)
Cognitive function	Normal (≥24)	170 (87.2)
	Mild impairment (20–23)	22 (11.3)
	Severe impairment (≤19)	3 (1.5)

**Table 2 ijerph-18-01688-t002:** Subjective evaluation of time management (*N* = 195).

Variable	Time Use: Min/Day Mean (SD)	Competency Mean (SD)	Importance Mean (SD)	Enjoyment Mean (SD)	Sample Range Mean (SD)	Evaluation Total Mean (SD)
Work	52.15 (112.49)	4.68 (0.63)	4.70 (0.62)	4.77 (0.48)	0 to 5	14.15(1.52)
Daily living activities	174.62 (90.79)	3.45 (0.90)	3.54 (0.79)	3.43 (0.81)	1 to 5	10.42 (2.05)
Leisure	416.92 (151.48)	3.71 (0.95)	3.87 (0.89)	3.98 (0.80)	1 to 5	11.56(2.30)
Rest	590.92 (105.12)	3.19 (1.30)	3.95 (0.85)	3.67 (0.85)	1 to 5	10.81(2.38)
Health management	205.38 (108.50)	4.24 (1.01)	4.72 (061)	4.64 (0.62)	0 to 5	13.62(1.88)

**Table 3 ijerph-18-01688-t003:** Type of health management activities.

Category		Min/Day Mean (SD)
Self-treatment	Medication	46.31 (31.39)
	Use of massage chair	1.08 (9.33)
	Complementary therapy	1.39 (15.66)
Health service	Orthodox medical service use	3.56 (22.05)
	Other health service use	3.56 (18.61)
Physical activity	Physical therapy or rehabilitation	4.97 (20.39)
	Exercise	136.55 (96.09)
Other health-related activities	Acquiring health information	1.24 (10.50)
	Resting when ill	4.79 (21.81)
	Monitoring health status	2.63 (11.33)

**Table 4 ijerph-18-01688-t004:** Health status and health management time (*N* = 195).

Variables		N (%)	Health Management Time		
			Mean (SD)	χ²	*p*-Value
Chronic disease	Total Mean (SD)	2.36 (1.36)			
	Yes	183 (93.8)	209.84 (108.01)	7.354	0.007
	No	12 (6.2)	137.50 (9.12)		
Medication	Total Mean (SD)	2.37 (1.99)			
	Yes	170 (87.2)	214.24 (109.19)	10.037	0.002
	No	25 (12.8)	145.20 (82.92)		
Regular exercise	Yes	177 (90.8)	214.41 (106.84)	15.48	<0.001
	No	18 (9.2)	116.67 (83.53)		
Blood pressure	Normal (<120/<80)	22 (11.3)	208.64 (83.57)	1.15	0.765
	Prehypertension (120–139/80–89)	82 (42.1)	206.34 (115.99)		
	Hypertension, stage 1 (140–159/90–99)	74 (37.9)	198.24 (98.67)		
	Hypertension, stage 2 (≥160/≥100)	17 (8.7)	227.65(142.33)		
Fasting glucose	Normal ^a^ (FBS < 140)	131 (67.2)	186.18 (96.15)	11.503 A z< b	0.003
	Low risk ^b^ (FBS: 140~199)	44 (22.6)	254.32 (131.88)		
	High risk ^c^ (FBS ≥ 200mg/dl)	20 (10.3)	223.50 (95.60)		
Diabetic	Well controlled ^a^ (HbA1C < 7)	23 (11.8)	223.04 (120.28)	8.248 a < b	<0.001
	Poorly controlled ^b^ (HbA1C ≥ 7)	44 (22.6)	257.05 (119.02)		
Lipid	Normal ^a^ (LDL < 129)	174 (89.2)	211.38 (109.28)	7.933	0.019
	Low risk ^b^ (LDL = 130~149)	15 (7.7)	172.00 (96.53)		
	High risk ^c^ (LDL ≥ 150)	6 (3.1)	115.00 (55.05)		
Nutritional status	Good (0–2)	103 (52.8)	221.36 (110.28)	7.500	0.024
	Moderate risk (3–5)	89 (45.6)	185.06 (104.83)		
	High risk (≥6)	3 (1.5)	260.00 (45.83)		
Depression	Normal ^a^ (0~4)	86 (44.1)	214.88 (112.99)	1.496	0.473
	Moderately depressed ^b^ (5~9)	75 (38.5)	191.60 (91.06)		
	Depressed (10~15)	34 (17.4)	211.76 (130.44)		
Frailty	Robust (0–2)	100 (51.3)	205.50 (111.55)	0.807	0.668
	Pre-frailty (3–4)	59 (30.8)	206.95 (101.15)		
	Frailty (≥5)	36 (17.9)	202.50 (114.38)		

Notes: FBS = fasting blood sugar; HbA1C = glycated hemoglobin; LDL= low-density lipoprotein; ^a, b,^ and ^c^ = Mann Whitney *U* test.

**Table 5 ijerph-18-01688-t005:** Factors related to the life satisfaction of older adults.

Predictors	Criterion: Health Management Time Likelihood χ² = 40.61, *p* < 0.001.			
	*β*	SE	Wald χ²	*p*
(Constant)	4.606	0.385	143.455	<0.001
Do not exercise	−0.489	0.124	15.549	<0.001
Number of chronic diseases	0.090	0.039	5.418	0.020
Number of medications	−0.023	0.026	0.742	0.389
FBS	0.001	0.001	4.411	0.036
LDL	-0.003	0.001	5.478	0.019
Nutrition	-0.035	0.023	2.333	0.127
Cognitive function	0.027	0.013	3.944	0.047
**Predictors**	**Criterion: Subjective Evaluation of Health Management** **Likelihood χ² = 21.48, p < 0.001.**			
	***β***	**SE**	**Wald χ²**	***p***
(Constant)	2.574	0.028	8200.545	<0.001
Do not exercise	−0.075	0.037	4.014	0.045
Health management time	0.000	0.000	4.528	0.033
Frailty	−0.016	0.006	7.650	0.006
Number of chronic diseases	0.018	0.009	4.254	0.039

Note: *N =* 195; FBS= fasting blood sugar; LDL = low-density lipoprotein; *β* = standardized coefficient; SE = standard error; VIF = variance inflation factor.

## Data Availability

Not applicable.

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
