# Peer review of "Daily Time Use by Activity of Community-Dwelling Older Koreans: Focus on Health Management"

_ijerph, 2021, doi:10.3390/ijerph18041688_

Round 1
Reviewer 1 Report
I believe that the problem of the article is relevant. The manuscript is organized in a clear and objective manner, and its presentation complies wirh the guidelines of IJERPH. I point out one or another aspect to consider, namely in the abstract, introduction and a typo at the beginning of the results.

Author Response
1. Abstract :
The abstract complies with IJERPH standards, in terms of word limit. The objetive of the study is not described in the abstract.
-> I clarified the aim of this study at the abstract (in red: Line 7 of the manuscript file).
This study aimed to examine the daily time use by activity and identified factors related to health management time (HMT) use among 195 older adults (mean age = 77.5, SD = 6.28 years; 70.8% women) attending a Korean senior center.
2. Key Words:
The keywords are representative of the subject studied and exposed and appear on the MeSH and DeCS platform
-> I searched the key words on the MeSH platform (www.nlm.nih.gov/mesh/MBrowser.html). So I changed the key words.
Health status, senior centers, time management, aged (in red:
Line 17 of the manuscript file)
3. Introduction:
The introduction shows the state of the art in relation to the study and is well founded. The objectives of the study are well defined here. The general objective of the research must be clarified. It would be important to justify the choice and importance of studying this theme, supporting it.
-> I revised the introduction section as suggested, providing more details. (in red: Line 98~100 of the manuscript file)
It aimed to 1) understand older adults’ general characteristics and subjective evaluation of time management, 2) explore health management activities, and the differences of HMT in relation to health status, and 3) identify factors influencing HMT and SEHM.
4. Results:
The results are displayed in a clear and attractive way, the tables have an easy interpretation of the results. In line 184 they start the sentence saying “The text continues here”, I believe that this statement is out of context, check !!
-> Thank you for your kindly reviews. I deleted that sentence. (Line 194 of the manuscript file)
Reviewer 2 Report
This study examined daily time use, analyzed the specific activities performed by older adults for health management, and identified influential factors. The study was conducted to provide basic data for the development of effective programs and health management activities for older adults attending senior centers. The first aim is to understand older adults’ daily time use on health management, health status, and subjective evaluation of health management and the second is to identify factors influencing HMT and SEHM
I feel it is a well-written, well-structured manuscript, and I outline some suggestions for authors to consider
Introduction,
It is well writen, please check if the first objective as written has been achieved after reading the results.
Methods,
I consider this an analytical study, not a descriptive study. Could the authors please review the study design?
Discussion,
Minior, please review line 231 se-nior
Some aspects of the discussion could go to the Introduction, as they present the state of the subject but do not discuss results.
All the best in your submission!
Author Response
I appreciate the time and effort spent by the reviewers to provide feedback and helpful suggestions for improving the article. I carefully considered the reviewers’ comments and made numerous edits to the manuscript and highlights. I responded to each of the reviewers’ comments and outlined my manuscript revisions in the included Response to Reviewers document. I am hopeful that the revised manuscript will be worthy of publication in International Journal of Environmental Research and Public Health. I thank the editor and the reviewers for the thoughtful comments and suggestions.
Please see the attachment.

Reviewer 3 Report
The article presents an interesting topic on the use of life time.Even so, possible changes:
-Introduction, tables and conclusions, exercise is confused with physical
activity, which is not the same, and must be corrected
-Expand the introduction, better contextualizing the situation,
with data from national and international organizations
-Justify What is the importance of social health in this situation,
the reason for the investigation.
-Incorporate a theory or support model that this article has.
-Methodology:
Indicate index of reliability, validity,
cronbach's alpha of each questionnaire and a citation of a recent
study that uses it in older people at the end of each questionnaire.
-Conclusion:
What does your study contribute that others do not
contribute?
What has limitations?
What theoretical implications does this article entail?
What practical implications for the elderly does this article entail?
What scientific implications do they suppose for other researchers?
Author Response

(The authors gave the same response as above.)

Round 2
Reviewer 2 Report
The manuscript has been improved, it can be considered by the editor for acceptance.
Reviewer 3 Report
Dear author,
Suggestions for improvement have been incorporated into the article and it has been improved for publication.
Best regards,